# Molecular insights into substrate recognition and discrimination by the N-terminal domain of Lon AAA+ protease

**Shiou-Ru Tzeng[1]\*, Yin-Chu Tseng[1], Chien-Chu Lin[2†], Chia-Ying Hsu[1], Shing-Jong Huang[3], Yi-Ting Kuo[1], Chung-I Chang[2,4]\***

[1]Institute of Biochemistry and Molecular Biology, College of Medicine, National Taiwan University, Taipei, Taiwan; [2]Institute of Biological Chemistry, Academia Sinica, Taipei, Taiwan; [3]Instrumentation Center, National Taiwan University, Taipei, Taiwan; [4]Institute of Biochemical Sciences, College of Life Science, National Taiwan University, Taipei, Taiwan

**Abstract** The Lon AAA+ protease (LonA) is a ubiquitous ATP-dependent proteolytic machine, which selectively degrades damaged proteins or native proteins carrying exposed motifs (degrons). Here we characterize the structural basis for substrate recognition and discrimination by the N-terminal domain (NTD) of LonA. The results reveal that the six NTDs are attached to the hexameric LonA chamber by flexible linkers such that the formers tumble independently of the latter. Further spectral analyses show that the NTD selectively interacts with unfolded proteins, protein aggregates, and degron-tagged proteins by two hydrophobic patches of its N-lobe, but not intrinsically disordered substrate, α-casein. Moreover, the NTD selectively binds to protein substrates when they are thermally induced to adopt unfolded conformations. Collectively, our findings demonstrate that NTDs enable LonA to perform protein quality control to selectively degrade proteins in damaged states and suggest that substrate discrimination and selective degradation by LonA are mediated by multiple NTD interactions.

**\*For correspondence:**
srtzeng@ntu.edu.tw (S-RT);
chungi@gate.sinica.edu.tw (C-IC)

**Present address:** [†]Division of Chemical Biology and Medicinal Chemistry, Eshelman School of Pharmacy, University of North Carolina at Chapel Hill, Chapel Hill, NC, United States

**Competing interests:** The authors declare that no competing interests exist.

## Introduction

The Lon AAA+ protease (LonA), previously known as the protease La, is an ATP-dependent protease distributed in prokaryotes and eukaryotes (*Charette et al., 1981*). It forms a homo-hexamer to execute its biological function (*Park et al., 2006*; *Goldberg et al., 1994*). LonA belongs to the AAA+ (ATPases associated with various cellular activities) superfamily and contains an N-terminal domain (NTD), a middle ATPase domain with conserved Walker motifs for ATP hydrolysis, and a C-terminal protease domain (CTD) with a serine-lysine catalytic dyad in the active site (*Rotanova et al., 2004*; *Botos et al., 2004*). LonA is responsible for degrading damaged or unfolded proteins, as well as native proteins bearing specific recognition elements, known as degradation tags or degrons (*Higashitani et al., 1997*; *Wohlever et al., 2014*). How can LonA discriminate its substrates from other non-substrate proteins in a cell? Substrate recognition of LonA is thought to be mediated by its NTD (*Ebel et al., 1999*; *Roudiak and Shrader, 1998*; *Rudyak and Shrader, 2000*; *Iyer et al., 2004*; *Melnikov et al., 2008*; *Adam et al., 2012*; *Cheng et al., 2012*). The substrates are unfolded and translocated in an ATP-dependent process into a secluded chamber formed by the ATPase and protease domains. Finally, the unfolded substrates inside the chamber can be degraded into small peptide fragments (*Gottesman, 2003*; *Baker and Sauer, 2006*). The ATP-dependent translocation process of substrates into the hexameric chamber has been well characterized (*Lin et al., 2016*; *Su et al., 2016*). Structures of different N-terminal fragments of LonA had been reported (*Li et al., 2005*; *Duman and Löwe, 2010*; *Li et al., 2010*; *Chen et al., 2019*), including *Escherichia coli*

**eLife digest** There are many different types of protein which each have different roles in biology. Most proteins are surrounded by water and are folded so that their water-attracting regions are on the outside and more fat-like regions, which repel water, are on the inside. When a protein becomes damaged or is assembled incorrectly, some of the fat-like regions end up on the outside of the protein and become exposed to water. This can prevent the protein from performing its role and harm the cell instead.

LonA proteases are responsible for dismantling and recycling these harmful proteins, as well as proteins that have been labelled for destruction. They do this by unfolding the unwanted protein and transporting it into an enclosed chamber made of six LonA molecules. Once inside the chamber, the target protein is broken down into smaller fragments that can be used to build other structures.

LonA proteases contain a region called the N-terminal domain, or NTD for short, which is thought to be responsible for identifying which proteins need degrading. Yet it remained unclear how the NTD recognizes and binds to these target proteins. To answer this question, Tzeng et al. studied the detailed structure of a LonA protease that had been purified from bacteria cells. This revealed that the NTD of LonA contains two water-repelling regions which bind to fat-like segments on the surface of proteins that have become unfolded or tagged for destruction.

Further experiments showed that the NTD is bound to the main body of LonA via a 'flexible linker'. This led Tzeng et al. to propose that the NTD sways around loosely at the end of LonA searching for proteins with exposed water-repelling regions. Once an NTD identifies and attaches to a target, the NTDs of the other LonA molecules then bind to the protein and help insert it into the chamber.

Proteases are a vital component of all biological systems. Controlling protein destruction and recycling is a key factor in how cells divide and respond to a changing environment. This study provides new insights into how LonA operates in bacteria, which may apply to proteases more widely. This contributes to our knowledge of fundamental biology and may also be relevant in a range of diseases where protein recycling is defective or inefficient.

(EcLon), *Mycobacterium avium complex* (MacLon), and *Bacillus subtilis* (BsLon). However, no structural study has been conducted to analyze substrate interactions by the NTD either in isolation or in the context of full-length LonA.

Here we address the question of substrate recognition and discrimination by the NTD of LonA using nuclear magnetic resonance (NMR) spectroscopy. To understand how LonA selectively recognizes protein substrates being in damaged states, we have used a thermal stable LonA isolated from *Meiothermus taiwanensis* (termed MtaLonA henceforth), which allows temperature cycling experiments to be used for switching the populations of protein substrates in the folded and unfolded states. In this work, five various protein substrates have been employed for NMR experiments, which include (1) Ig2 (domains 5 and 6 of the gelation factor from *Dictyostelium discoideum* [*Hsu et al., 2009*], hereafter abbreviated as Ig2) with an immunoglobulin (Ig) fold forming a dimer that is natively folded up to 40°C; (2) α-casein, which is an intrinsically disordered protein with no tertiary structure; (3) native lysozyme with four disulfide bridges and reduced lysozyme forming loose and flexible aggregates; (4) Ig2D5 (domain 5 of the gelation factor from *D. discoideum*), which is used for characterization of substrate conformation selection by the NTD; and (5) a degron-tagged Ig2D5 designed to identify the residues of degrons that can be recognized and bound by the NTD.

Our results show that the presence of MtaLonA NTD is required to degrade damaged proteins and native proteins with degrons, but it does not play an active role in mediating the degradation by LonA of an intrinsically disordered substrate, α-casein. Here we report the crystal structure of an N-terminal fragment of MtaLonA determined at a 2.1 Å resolution, demonstrating that it is similar to the structures of both EcLon and MacLon NTDs, but not to that of BsLon. MtaLonA NTD appears to tumble independently of the hexameric core complex, indicating that the NTD is attached to the hexameric chamber by a flexible linker thus rendering it possible for detailed chemical shift perturbation (CSP) mapping of substrate binding in the context of full-length MtaLonA. We further

structurally characterize the NTD-mediated interactions with unfolded proteins, protein aggregates, and degron-tagged proteins. This work suggests that the flexibly linked NTDs can help survey, discriminate, and selectively capture damaged unfolded protein species or native protein substrates carrying exposed degradation sequence motifs.

## Results

### The NTD is essential for efficient degradation of a thermally unfolded substrate but not intrinsically disordered protein substrate, α-casein

To evaluate the importance of the NTD in the proteolysis activity of MtaLonA, we constructed a Mta-LonA variant, AAAP, of which the NTD (residues 1–241) was removed (*Figure 1A*), based on the structure of *B. subtilis* LonA. Substrate degradation activity of AAAP was evaluated using two protein substrates, namely α-casein and Ig2. Native α-casein is an intrinsically disordered substrate of LonA. Ig2 is an all β-stranded protein that becomes partially unfolded at 55°C (*Figure 1—figure supplement 1*) and only then is susceptible to proteolysis by MtaLonA (*Higashitani et al., 1997*). Full-length MtaLonA degrades both α-casein and damaged Ig2 (*Van Melderen et al., 1996*). Compared to the full-length MtaLonA, AAAP exhibited a slightly reduced degradation activity against α-casein (*Figure 1B*), while its proteolytic activity for thermally unfolded Ig2 was severely impaired (*Figure 1C*). These results showed that, despite the lack of the NTD, AAAP retains the proteolytic activity for intrinsically disordered protein substrates like α-casein. However, the NTD is essential for MtaLonA to recognize and degrade the thermally damaged substrate, Ig2, indicating the NTD may mediate specific interactions with damaged Ig2, but not α-casein.

### The NTDs tumble independently of the hexameric LonA core in solution

We obtained crystals of the N-terminal fragment NN206 of MtaLonA by in situ proteolysis during crystallization (*Figure 2A*). The structure was refined to 2.1 Å resolution with a free R factor of 24.9% (crystallographic R factor is 20.9%; *Table 1*). NN206 forms a bilobal fold: (1) the N-terminal lobe, which consists of one α-helix and six β-strands, forming three β-sheets (β1/β3/β4/β5, β2/β5/β4, and β1/β6/β5); (2) the C-terminal lobe, which is exclusively α-helical, consisting of a five-helix bundle (*Figure 2B*). The two lobes are jointed together by an 18-residue linker, which is similar to that of both EcLon and MacLon (*Li et al., 2005*; *Duman and Löwe, 2010*; *Li et al., 2010*; *Chen et al., 2019*). Based on the result, we also crystallized and solved the structure of the longer N-terminal fragment NN291 (*Figure 2A*). The structure of NN291, refined to 1.7 Å resolution (*Table 1*), is very similar to that of NN206 except for a longer C-terminal helix by four more residues, that is, residues 207–210 (*Figure 2C*). Despite the lack of spontaneous proteolytic fragmentation in the crystals, no electron density was seen past residue A210 (*Figure 2D*).

Interestingly, the C-terminus of *E. coli* N-terminal fragment forms an extended 40-residue helix (*Li et al., 2010*). Therefore, we further investigated the structural feature of the corresponding residues 207–243 of MtaLonA using solution-state NMR spectroscopy. Most of the well-dispersed cross-peaks in two-dimensional $^{15}$N-$^{1}$H HSQC NMR spectra of NN206 and NN243 can be well superimposed, indicating that the folded domain structure corresponding to residues 1–206 is not perturbed by the C-terminal extension, residues 207–243 (*Figure 2—figure supplement 1A*). Nonetheless, NN243 exhibited extra intense and poorly dispersed cross-peaks around 7.5–8.5 ppm along the proton dimension that is indicative of a highly disordered peptide segment, which most likely corresponds to the C-terminal extension of NN243. Together, our crystallographic and NMR analyses suggest that the NTD of MtaLonA forms a bilobal globular structure (residues 1–206) followed by a C-terminal flexible extension (residues 211–243).

LonA forms a large homo-hexameric complex resulting in a very high molecular weight of 0.5 MDa. To verify the dynamic feature of the NTD, we compared the backbone amide $^{15}$N-$^{1}$H NMR correlations of protonated NN206 and those of full-length MtaLonA (with perdeuteration of the non-exchangeable hydrogens) recorded at 55°C. With the aid of perdeuteration labeling and transverse relaxation optimized spectroscopy (TROSY), we observed over 200 well-resolved $^{15}$N-$^{1}$H correlations of full-length hexameric MtaLonA at an apparent molecular weight of 0.5 MDa. Importantly, most of these correlations were very similar to those observed in the NN206 spectrum (*Figure 2E*) and a systematic chemical shift offset was due to deuterium isotope effects. These results indicated that the

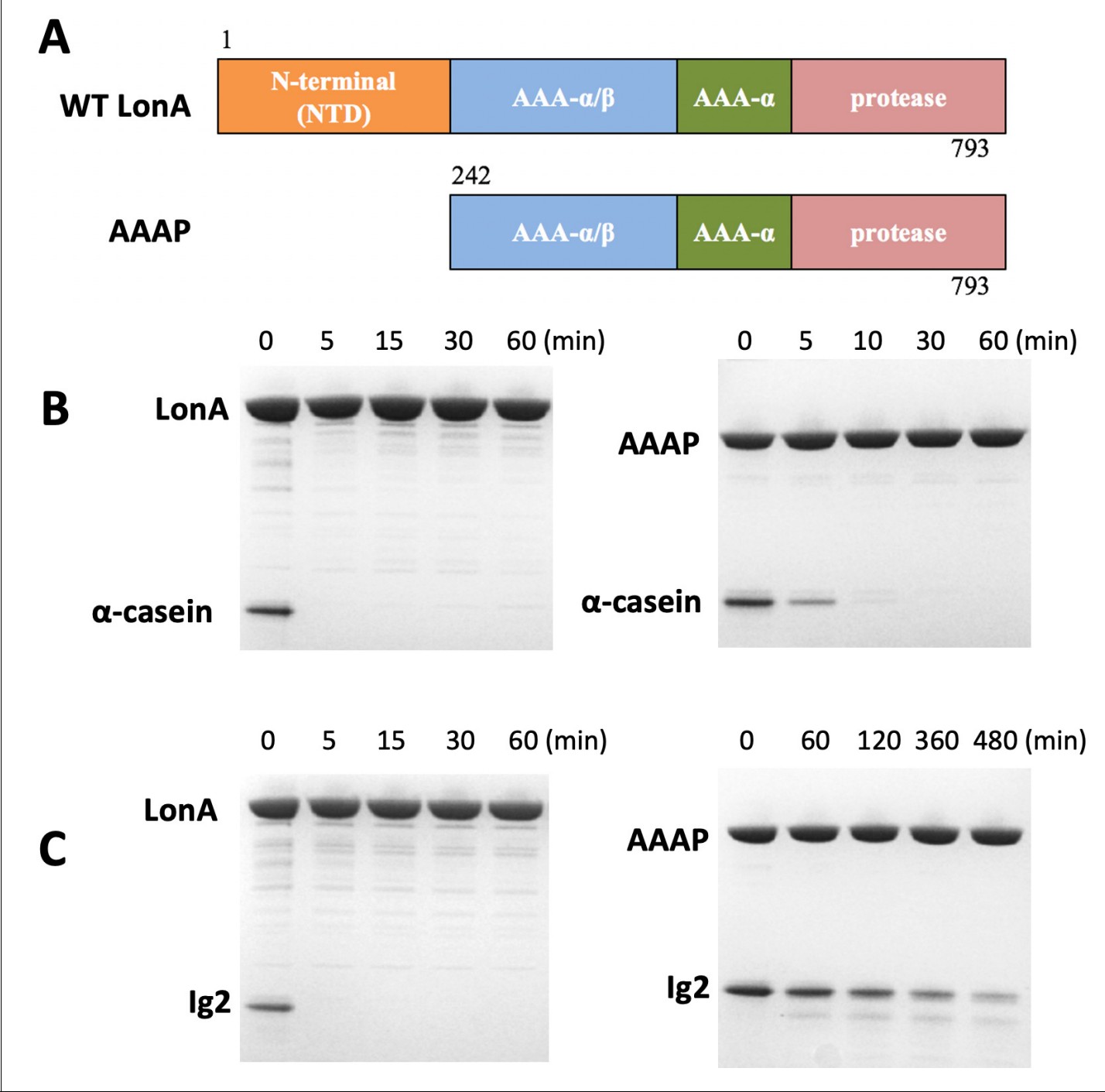

**Figure 1.** The NTD of *Meiothermus taiwanensis* LonA (MtaLonA) is essential for efficient degradation of a thermally unfolded substrate, but not the intrinsically disordered substrate, α-casein. (A) Schematic diagrams illustrating domain organization of full-length MtaLonA and the N-terminal domain truncated AAAP. (B) Degradation of 4 µM α-casein by 0.4 µM (hexamer) of full-length MtaLonA (FL) and AAAP at 55˚C. (C) Degradation of 4 µM Ig2 by 0.4 µM (hexamer) of full length and AAAP at 55˚C.

The online version of this article includes the following figure supplement(s) for figure 1:

**Figure supplement 1.** The thermal stability of LonA, Ig2, and NN206.

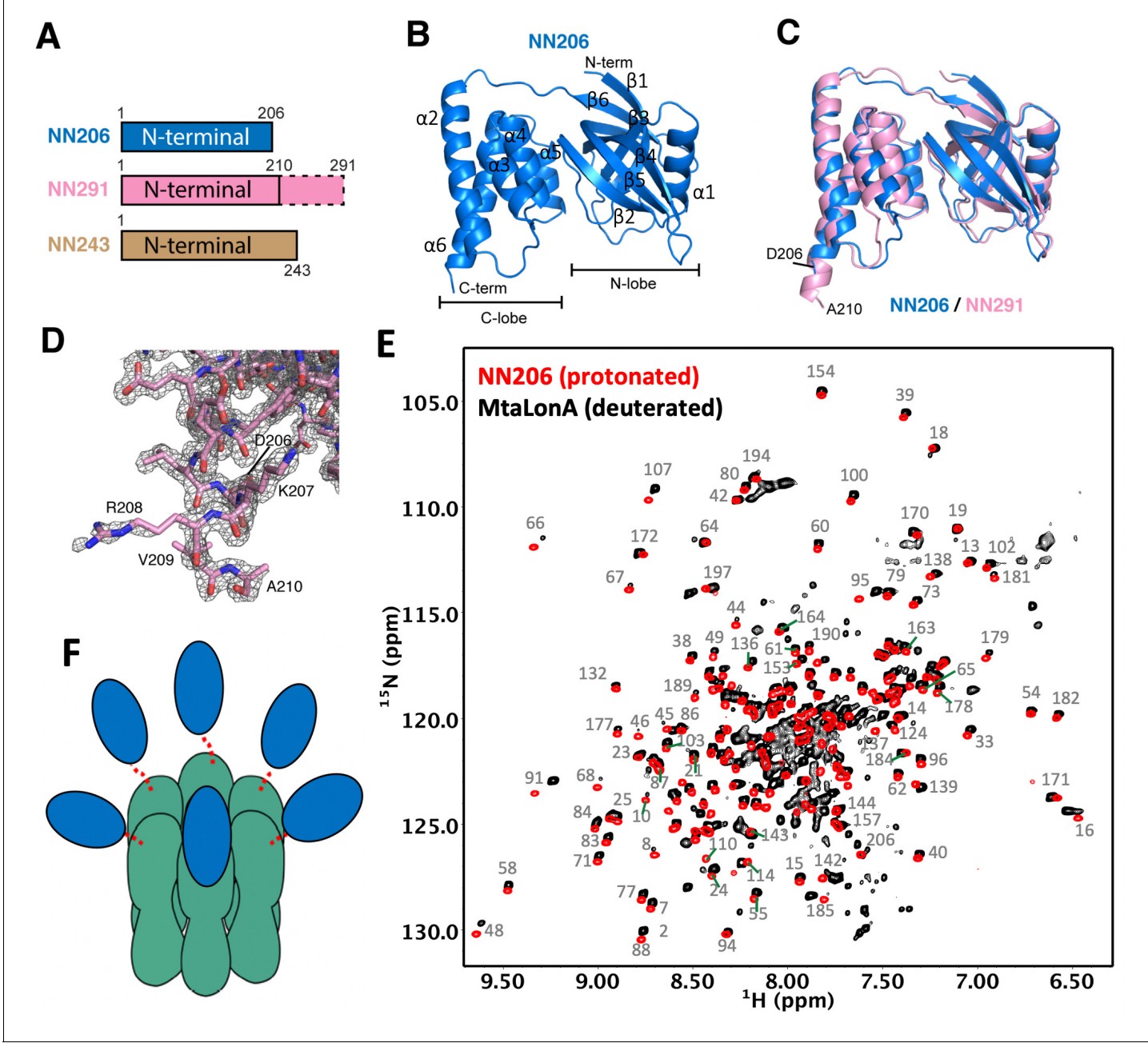

**Figure 2.** Overall structures of the N-terminal domain of MtaLonA. (A) The three N-terminal constructs NN206, NN291, and NN243. The disordered region is shown in the dashed box. (B) Structure of NN206. NN206 forms a bilobal fold: (1) the N-terminal lobe forming three β-sheets (β1/β3/β4/β5, β2/β5/β4, and β1/β6/β5); (2) the C-terminal lobe consisting of a five-helix bundle. (C) Superposition of NN206 (marine) and NN291 (pink) structures. (D) The refined 2Fo-Fc electron density map around the C-terminal residue A210 of NN291, contoured at 1.0 $\sigma$. (E) Comparison of $^1$H–$^{15}$N TROSY-HSQC spectra of protonated NN206 (red) and highly deuterated$^{15}$N-labled MtaLonA (black) recorded at 55°C. A systematic chemical shift offset is due to deuterium isotope effects. The well-dispersed resonances are labeled with residue numbers. About 200 well-resolved correlations are superimposable, suggesting that NTD is loosely linked to the hexameric core via a flexible linker. (F) Each of the six NTDs may connect to the hexameric core of fused AAA+ and protease domains by a flexible ~40-residue linker. The NTDs and ATPase–Protease chamber are illustrated in blue and green colors, respectively. Dashed lines represent the flexible linker regions.

The online version of this article includes the following figure supplement(s) for figure 2:

**Figure supplement 1.** NMR characterization of MtaLonA NTD.

**Table 1.** Data collection and refinement statistics.

| | NN206 | NN291 |
|---|---|---|
| PDB entry | 7CR9 | 7CRA |
| **Data collection** | | |
| Space group | $P2_12_12_1$ | $P6_322$ |
| **Cell dimensions** | | |
| a, b, c (Å) | 32.830, 58.296, 198.492 | 86.016, 86.016, 110.126 |
| α, β, γ (°) | 90, 90, 90 | 90, 90, 120 |
| Resolution (Å) | 30–2.09 (2.16–2.09)* | 30–1.7 (1.76–1.7) |
| $R_{merge}$ | 0.106 (0.937) | 0.069 (0.779) |
| $I/\sigma I$ | 17.9 (3.0) | 49.1 (4.7) |
| Completeness (%) | 99.7 (100) | 100 (100) |
| Redundancy | 6.5 (6.6) | 20.9 (21.4) |
| **Refinement** | | |
| Resolution (Å) | 29.409–2.1 (2.149–2.1) | 30–1.7 (1.744–1.7) |
| $R_{work}/R_{free}$ | 0.209/0.249 | 0.197/0.237 |
| **No. atoms** | | |
| Protein<br>Ligand ($SO_4^{2-}$) | 3285<br>0 | 1684<br>10 |
| Water | 107 | 244 |
| **B-factors** | | |
| Protein<br>Ligand ($SO_4^{2-}$) | 53.14 | 16.18<br>27.45 |
| Water | 52.66 | 28.21 |
| **R.m.s deviations** | | |
| Bond lengths (Å) | 0.004 | 0.009 |
| Bond angles (°) | 0.74 | 1.34 |
| **Ramachandran** | | |
| Favored (%) | 96.81 | 96.67 |
| Allowed (%) | 2.46 | 2.38 |
| Outliers (%) | 0.74 | 0.95 |

Number of crystals for each structure should be noted in footnote.

*Highest resolution shell is shown in parentheses.

NTD adopts the same structure as the isolated NTD while loosely tethered to the near-megadalton hexameric LonA core via a long flexible linker leading to an independently fast tumbling motion to yield highly favorable NMR relaxation properties and thus high-quality multidimensional NMR spectra (*Figure 2—figure supplement 1B and C*). We further applied multidimensional heteronuclear NMR experiments to assign the observed cross-peaks in $^1H$–$^{15}N$ TROSY-HSQC spectrum of NN206 (BMRB code: 50697) and the NMR-derived secondary structure of NN206, based on the Chemical Shift Index (*Hafsa et al., 2015*), was highly similar to its solved X-ray structures (*Figure 2—figure supplement 1D*). Together, these results suggested that in a MtaLonA complex, each of the six NTDs may connect to the hexameric core by a flexible ~40-residue linker. Therefore, each 24-kDa NTD may tumble independently of the 350-kDa hexameric core (*Figure 2F*).

## The chemical shifts of MtaLonA NTDs are perturbed by thermally unfolded proteins, aggregates, and degron tags

To directly observe substrate discrimination mediated by the NTD, we treated NMR samples by one thermal cycle with temperature increasing from 32℃ to 55℃ and then returning to 32℃. With temperature rising, the exposed hydrophobic areas of thermally unfolded Ig2 was increased, as

detected by SYPRO Orange (*Figure 1—figure supplement 1*), indicating that the population of Ig2 in unfolded states is increased by thermal denaturation. By contrast, thermal cycling of isolated NN206 exhibited no temperature effects on its protein structure monitored by NMR spectroscopy (*Figure 3—figure supplement 1*), which could be explained by the high melting temperatures ($T_m$) of full-length MtaLonA and NN206 (68.0°C and 85.5°C, respectively; *Figure 1—figure supplement 1*). Upon addition of a twofold molar excess of well-folded Ig2 at 32°C, no significant CSPs were observed (*Figure 3A*), where only the NN206 was isotopically enriched. By increasing temperature from 32°C to 55°C for the same NMR sample, dramatic variations in the 2D spectral features were detected and the resonances underwent a prominent decrease in intensity (*Figure 3B*), indicating that the NTD interacts with thermally unfolded Ig2. With temperature dropping from 55°C to 32°C, linewidth of the expected bound-state resonances was too broad to be detected in the presence of binding events (*Figure 3C*), suggesting a certain portion of Ig2 still stays in the unfolded or aggregated state. In contrast, the addition of α-casein resulted in little to no effect on the cross-peaks of residues 1–206 of the NTD (*Figure 3—figure supplement 2A and B*), thus precluding specific interaction mode between the intrinsically disordered protein α-casein and MtaLonA NTD.

To clarify whether overall broadening of the cross-peak signals may be due to protein aggregation or precipitation, we performed size-exclusion chromatography (SEC) experiments to examine the NMR sample treated with thermal cycling. The analysis revealed that more than half of Ig2 protein became unfolded or aggregated through one thermal cycle while the NTD was still well-folded (*Figure 3D*). The NTD-Ig2 aggregate complex showed a dissociable SEC profile, suggesting the NTD-substrate interaction is quite dynamic by nature and the fast substrate dissociation may facilitate the translocation process.

NMR analysis indicates that thermally damaged Ig2 induces significant CSPs and broadened resonances in the NTD N-lobe, which comprises one α-helix and three β-sheets β1/β3/β4/β5, β2/β5/β4 and β1/β6/β5. The residues of MtaLonA NTD with the pronounced CSPs are part of the interaction interface located mainly at helix α1, loop L1, L2, L3, and β-sheet β2/β5/β4 (*Figure 3E* and *Figure 3—figure supplements 3* and *4A*), which consist primarily of hydrophobic residues and are decorated by a number of polar residues (*Figure 3—figure supplement 4B and C*). To characterize the binding sites of MtaLonA NTD interaction with damaged Ig2, we chose to mutate two exposed hydrophobic residues P22 and M85 located at the center of β-sheet β2/β5/β4 (*Figure 3—figure supplement 4B*). We found that NN206*, containing two specific mutations P22A and M85A (hereafter NN206*), may influence the substrate-binding interaction. Spectral analysis showed chemical shift differences between NN206 and NN206* can be negligible (*Figure 3—figure supplement 4D and E*), suggesting that they have very similar structures, with the exception of the region surrounding the point substitutions. The NMR sample containing both [15]N-labeled NN206* and unlabeled Ig2 was also treated with one thermal cycle. NN206* showed much weak interactions with thermal damaged Ig2 (*Figure 3F–H* and *Figure 3—figure supplement 4F*), confirming that this binding event is mainly contributed by the specific hydrophobic residues with high hydrophobicity in the N-lobe. Collectively, our results showed that the NTD of LonA selectively interacts with protein substrates via critical hydrophobic residues of its N-lobe, demonstrating that the NTD enables LonA to perform protein quality control by selectively interacting with proteins in damaged or unfolded states.

To further examine how the NTD is involved with protein aggregation, we probed the interactions between the scrambled lysozyme aggregate and MtaLonA NTD by NMR spectroscopy. Native lysozyme is a 14.3-kDa protein (129 amino acids) comprising four disulfide bridges while reduced lysozyme forms loose and flexible aggregates (*Yang et al., 2015*). TCEP (Tris (2-carboxyethyl) phosphine; nonthiol-based reducing agent) drives the formation of amorphous lysozyme aggregates. Moreover, there is no cysteine residue found in full-length MtaLonA, suggesting TCEP does not affect the structure of MtaLonA NTD. Addition of TCEP-treated denatured lysozyme leaded to substantial spectral changes of NN206 resonances recorded at 32°C, while NN206* only showed marginal CSPs (*Figure 3—figure supplement 5A*). The structural mapping showed that the affected residues were also located mainly in the N-lobe shown in *Figure 3—figure supplement 5B and C*. Furthermore, we also examined the degradation activity of MtaLonA against either native or denatured lysozyme, respectively. A gel-based assay showed that full-length MtaLonA can efficiently degrade denatured-reduced lysozyme, but not native lysozyme (*Figure 3—figure supplement 5D*). Accordingly, AAAP, without the NTD, was inactive against either the scrambled lysozyme aggregate or native lysozyme (*Figure 3—figure supplement 5E*). Together, these results highlighted the key

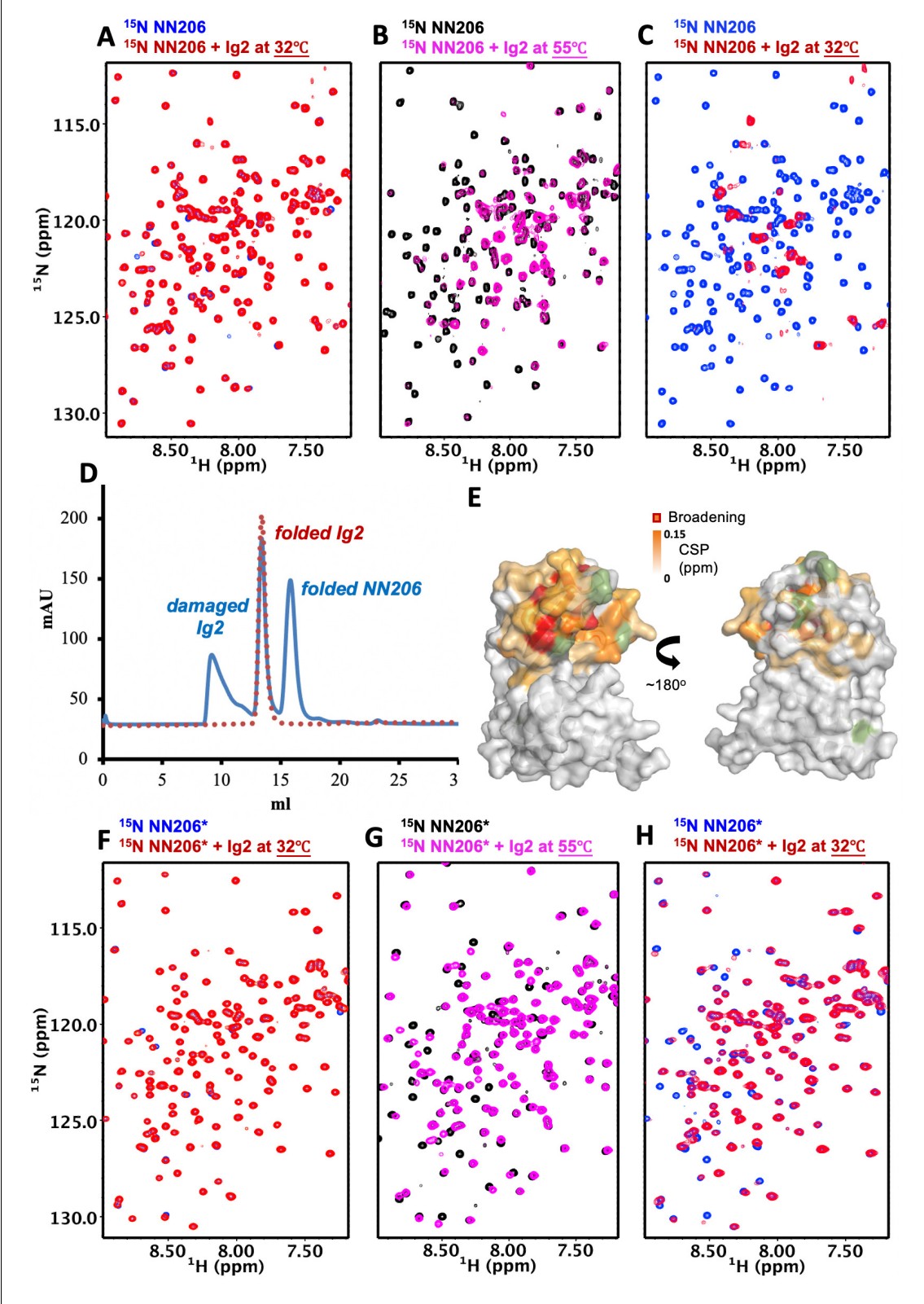

**Figure 3.** The N-lobe of MtaLonA NTD is involved in recognition of thermally unfolded substrate Ig2. (**A**) $^1$H–$^{15}$N TROSY-HSQC spectra of NN206 in the absence (blue) and presence (red) of unlabeled Ig2 recorded at 32°C. (**B**) Comparison of $^1$H–$^{15}$N TROSY-HSQC spectra of NN206 in the apo (black) and Ig2-bound (magenta) states recorded at 55°C. (**C**) After one thermal cycle, $^1$H–$^{15}$N TROSY-HSQC spectra of NN206 in the absence (blue) and presence (red) of Ig2 recorded at 32°C. (**D**) Gel filtration profiles of the NMR sample containing both $^{15}$N-labeled NN206 (300 µM) and unlabeled Ig2 (600 µM)

*Figure 3 continued on next page*

*Figure 3 continued*

treated with thermal cycling shown in blue and folded Ig2 (red dashed lines) analyzed by Superdex 200 10/300 GL column. (**E**) Interaction interface of NN206: Ig2 mapped onto the structure of MtaLonA NTD, based on the spectrum recorded at 55°C. Proline residue is shown in green. (**F**) $^1$H–$^{15}$N TROSY-HSQC spectra of NN206* in the absence (blue) and presence (red) of Ig2 recorded through thermal cycling from 32°C (**F**) to 55°C (**G**) and then returning to 32°C (**H**).

The online version of this article includes the following figure supplement(s) for figure 3:

**Figure supplement 1.** The thermal stability of NN206.
**Figure supplement 2.** NMR characterization of MtaLonA NTD interacting with α-casein.
**Figure supplement 3.** NMR analysis indicates that thermally damaged Ig2 induces significant CSPs and broadened resonances in the NTD N-lobe.
**Figure supplement 4.** NMR characterization of MtaLonA NTD* interacting with thermally damaged Ig2.
**Figure supplement 5.** Interaction of NTDs with the scrambled lysozyme.

role of NTD, via the N-lobe region, in discriminating and recognizing protein aggregates before subsequent translocation and degradation.

LonA can also degrade natively folded proteins with degradation tags such as Sul20 degron (*Higashitani et al., 1997*; *Gur and Sauer, 2008*). Previously, the NTD of *E. coli* LonA was shown to be required for degron binding by chemical cross-linking (*Wohlever et al., 2014*). To understand how the NTD directly interacts with degrons, a 20-residue peptide (the C-terminal 20 residues of SulA; namely Sul20) was synthesized and used for NMR titration experiments. Addition of unlabeled Sul20 caused significant spectral changes of NN206 resonances (*Figure 4A*) and the structural mapping showed that the affected residues were again located at β-sheet β2/β5/β4, loop L1, L3, and helix α1 of the N-lobe subdomain (*Figure 4B* and *Figure 4—figure supplement 1A*). Mutations of P22 and M85 located at the center of the hydrophobic surface to moderately hydrophobic Alanine leaded to much weak binding of NN206* with Sul20 peptide (*Figure 4—figure supplement 1B and C*), suggesting that specific hydrophobic residues of beta-stranded β2/β5/β4 located at the N-lobe of MtaLon NTD play a critical role in degron recognition.

To characterize the effect of MtaLon NTD on degrons further, we appended Sul20 to the C-terminus of a protein substrate Ig2D5 (domain 5 of the gelation factor from *D. discoideum*, hereafter abbreviated as Ig2D5) and investigated the conformational changes experienced by Sul20 upon binding the NTD. The NMR spectra of Ig2D5 and Ig2D5-S20 were well superimposed, indicating that the folded domain structure corresponding to Ig2D5 is not perturbed by the C-terminal Sul20 (*Figure 4C*). Furthermore, the NMR spectrum of Ig2D5-S20 exhibited additional cross-peaks that well corresponded to Sul20 and the NMR-derived secondary structure of Sul20 was flexible random-coil (*Figure 4—figure supplement 1D*). Measurement of the local backbone flexibility showed that elevated [$^1$H]–$^{15}$N nuclear Overhauser effect (NOE) values for most residues were from Ig2D5 (*Figure 4D*), along with the large increase in chemical shift dispersion. Lower [$^1$H]–$^{15}$N NOEs for Sul20 residues were due to the increased flexibility on ps-ns timescale. By addition of unlabeled NN206 to isotopically enriched Ig2D5-S20, we observed chemical shift changes and broadened resonances of C-terminal Sul20 (*Figure 4C and E*), suggesting that residues Y19 and H20 are strongly affected by the binding event based on the resonance broadening beyond detection. The secondary structure of Sul20 remained random coil in the presence of MtaLonA NTD (*Figure 4—figure supplement 1D*). The NMR spectrum of Ig2D5-S20 in the presence of an equimolar amount of unlabeled NN206* showed a much less pronounced chemical shift effect (*Figure 4—figure supplement 1E and F*) than wild-type NN206 binding to Ig2D5-S20, indicating that Sul20 tag can be recognized by the hydrophobic patches of N-lobe of MtaLon NTD. Furthermore, we generated a Y19A/H20A double mutant in Ig2D5-S20 construct (hereafter abbreviated as Ig2D5-S20-Y19A/H20A). To investigate an effect attributed to Y19A/H20A double mutant, a gel-based assay was performed at 42°C while Ig2D5 was still at its native state. The results showed that the degradation by the full-length MtaLonA against Y19A/H20A double mutant of Sul20 was much less efficient (*Figure 4—figure supplement 2A*). The NMR spectra of Ig2D5-S20 and Ig2D5-S20-Y19A/H20A were well superimposed and only the chemical shifts of residues Y19A and H20A were prominently changed (*Figure 4—figure supplement 2B*). By addition of $^{15}$N-labeled NN206 to $^{15}$N-labeled Ig2D5-S20-Y19A/H20A, careful comparison of chemical shifts revealed that no obvious changes, suggesting that Y19A/H20A double mutant indeed significantly reduces the binding with NN206 (*Figure 4F* and *Figure 4—figure supplement 2C*).

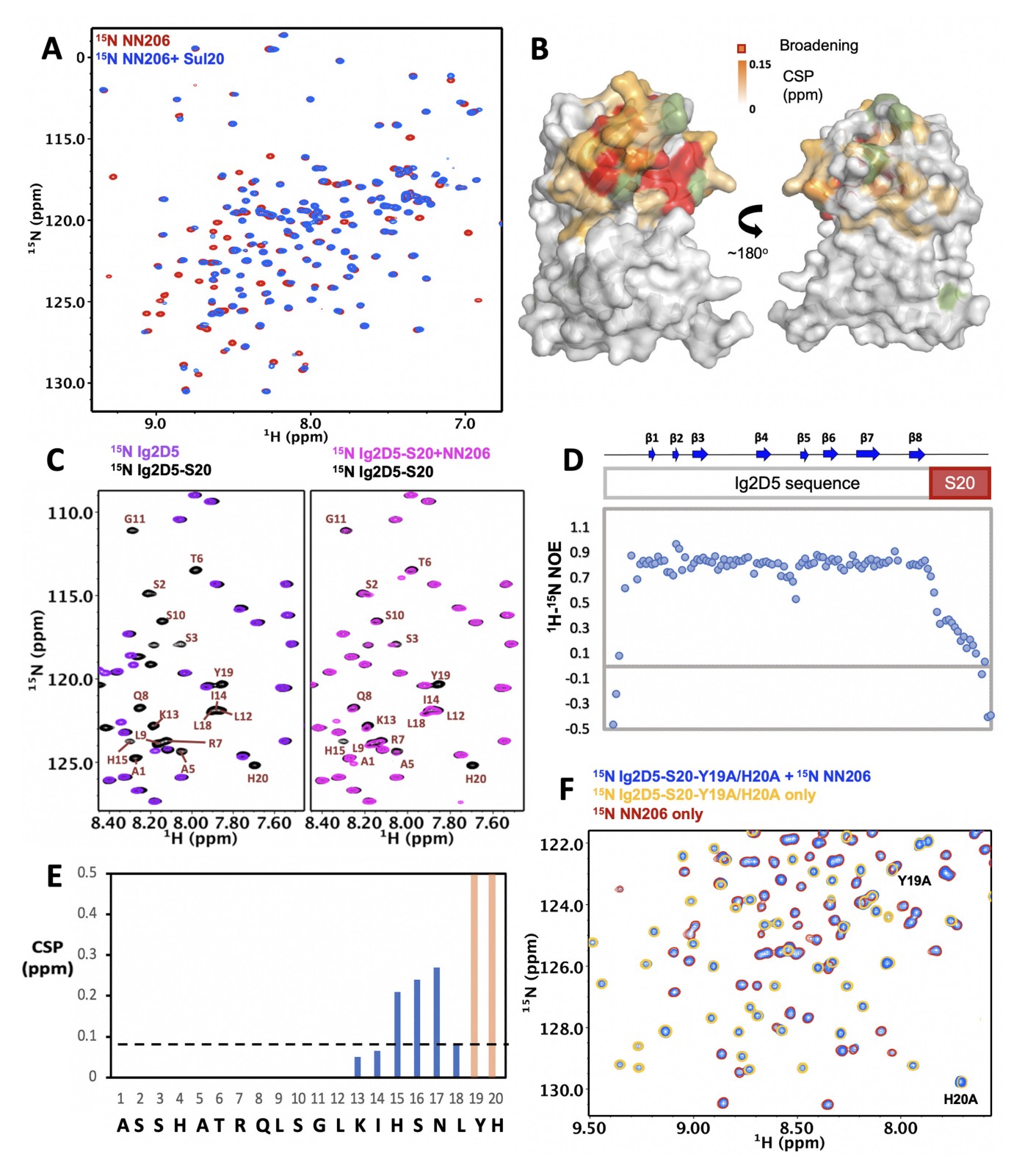

**Figure 4.** Interaction of MtaLonA NTDs with degron tag Sul20. (**A**) Overlay of $^1$H–$^{15}$N TROSY-HSQC spectra of NN206 in the absence (red) and presence (blue) of Sul20 peptide. (**B**) Structural mapping of the chemical shift perturbations (orange) of NN206 caused by Sul20 peptide. Proline residue is shown in green. (**C**) Overlay of $^1$H–$^{15}$N TROSY-HSQC spectra of Ig2D5 (purple) and Ig2D5-S20 (black); overlay of $^1$H–$^{15}$N TROSY-HSQC spectra of Ig2D5-S20 in the absence (black) and in the presence (pink) of unlabeled NN206. (**D**) [$^1$H]–$^{15}$N nuclear Overhauser effect (NOE) values of Ig2D5-S20. (**E**)

*Figure 4 continued on next page*

eLife Research article

Biochemistry and Chemical Biology | Structural Biology and Molecular Biophysics

*Figure 4 continued*

CSPs of amide moieties of Ig2D5-S20 after binding unlabeled NN206, plotted against Sul20 residue number. Residues Y19 and H20 are strongly affected by the binding event based on the resonance broadening beyond detection (shown in orange). A significance level for CSP is indicated by a dotted line. (F) Overlay of $^{1}$H–$^{15}$N TROSY-HSQC spectra of $^{15}$N-labeled NN206 in apo state (red) and $^{15}$N-labeled Ig2D5-S20-Y19A/H20A in the absence (yellow) and presence (blue) of $^{15}$N-labeled NN206.

The online version of this article includes the following figure supplement(s) for figure 4:

**Figure supplement 1.** Interaction of NN206* with degron tag Sul20.

**Figure supplement 2.** Interaction of Y19A/H20A double mutant of Sul20 tag with MtaLonA NTD.

## The thermally unfolded conformations of substrate proteins are stabilized by engagement with MtaLonA NTD

To understand further selective recognition of protein substrate in the unfolded states by the NTD, we asked whether the unfolding-refolding process of substrate may be modulated by interacting with the NTD. To investigate how the conformations of substrates are influenced in the presence of NTDs, we constructed a single-domain protein substrate Ig2D5 for NMR experiments due to its high-quality NMR spectrum (*Figure 5—figure supplement 1A*). About 96% of the backbone resonances of Ig2D5 can be assigned by multidimensional heteronuclear NMR experiments. Ig2D5 is a β-sheet protein with Ig-like fold and the $T_m$ of substrate Ig2D5 is similar to Ig2 (*Figure 5—figure supplement 1B and C*). Here we applied NMR spectroscopy to investigate the unfolding-refolding equilibrium of Ig2D5 induced by temperature cycling while the structure of MtaLon NTD remained stable. The NMR samples containing isotopically enriched Ig2D5 in the absence and presence of unlabeled NN206 were heated from 32°C to 60°C and then cooled to 32°C. Based on the intensity changes of resonances corresponding to native Ig2D5, we estimated that isolated Ig2D5 was largely unfolded at 60°C (*Figure 5A and B*; coloured green) and ~72% of Ig2D5 was folded after one thermal cycle (*Figure 5C* and *Figure 5—figure supplement 1D*). However, in the presence of equimolar NN206, the folded ratio of Ig2D5 was reduced to ~15% after thermal treatment (*Figure 5C*; coloured blue). Similar results were obtained using equimolar full-length MtaLonA with catalytic mutant S678A (LonA*) (*Figure 5C*; coloured orange). However, on addition of equimolar AAAP* (residues 242–793 with catalytic mutant S678A), ~75% of Ig2D5 were folded under the same condition (*Figure 5C*; coloured black). The results revealed that the unfolded state of Ig2D5 was engaged with MtaLon NTD and the temperature-induced unfolding-refolding of substrate was significantly affected in the presence of MtaLon NTD (*Figure 5D*). Accordingly, we also examined how the unfolding-refolding of Ig2D5 could be affected by NN206*. After one thermal cycle, about 75% of Ig2D5 was folded in the presence of equimolar NN206* (*Figure 5E* and *Figure 5—figure supplement 2A–C*). Upon addition of eightfold excess of NN206*, about 60% of Ig2D5 was folded (*Figure 5—figure supplement 2D*), suggesting that NN206* interacts with denatured protein substrates weakly. These results demonstrated that the conformations of Ig2D5 may be affected by the hydrophobic substrate-binding interaction and the thermally unfolded states of substrate proteins can be stabilized by engagement with the NTDs. Thus, by selectively interacting with hydrophobic residues exposed in the thermally unfolded states of substrate proteins, the NTDs prominently perturb temperature-induced unfolding-refolding process of substrates.

## Discussion

In this work, we used α-casein and Ig2, which represent two types of substrates. Native α-casein is an intrinsically disordered protein; by contrast, the native Ig2 is a β-sheet protein with $T_m$ at 46°C (*Figure 1—figure supplement 1*). We show that the construct AAAP devoid of the NTD (residues 242–793, ATPase and protease domains), which retains wild-type ATPase and peptidase activities in LonA, lacks the degradation activity against thermally denatured Ig2 while exhibits a wild-type-like activity against α-casein (*Figure 1*). Therefore, the NTD is specific for denatured or damaged protein substrates, whose hydrophobic core regions normally buried in the folded state become exposed, but not for intrinsically disordered substrates like casein, which contains mainly charged or polar residues and lack hydrophobic regions in the protein sequence. Indeed, it has been shown that LonA prefers substrates rich in hydrophobic residues (*Gur and Sauer, 2008*).

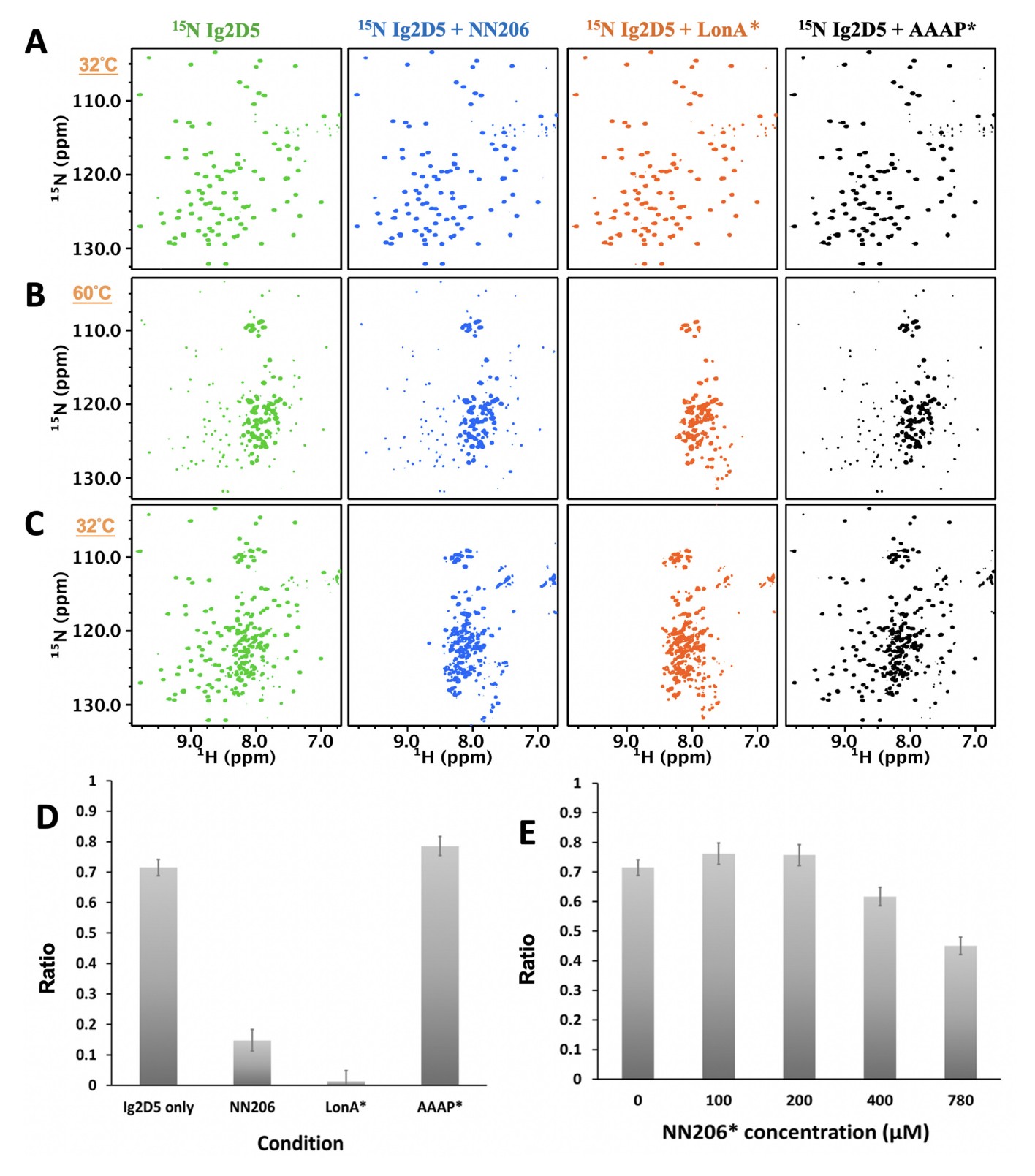

**Figure 5.** The unfolding-refolding of Ig2D5 is affected in the presence of MtaLonA NTDs. 2D $^1H$-$^{15}N$ TROSY-HSQC NMR spectra of Ig2D5 in the absence (green) and presence of NN206 (blue), LonA* (full-length MtaLonA with catalytic mutant S678A; coloured orange), or AAAP* (residues 242–793 with catalytic mutant S678A; coloured black) recorded through one thermal cycle starting from 32°C (**A**) to 60°C (**B**), and then returning to 32°C (**C**). (**D**)
*Figure 5 continued on next page*

*Figure 5 continued*

Folded ratios of Ig2D5 in the absence and presence of NN206, LonA* or AAAP* after one thermal cycle. (E) Folded ratios of apo Ig2D5 and Ig2D5 titrated with different concentrations of NN206* after one thermal cycle.

The online version of this article includes the following figure supplement(s) for figure 5:

**Figure supplement 1.** Biophysical characterization of Ig2D5.

**Figure supplement 2.** The unfolding-refolding of Ig2D5 in the presence of NN206*.

Our crystallographic results of NN206 and NN291 are consistent with previous findings that the ~40 residue region between the NTD and ATPase–Protease domains is structurally flexible with multiple accessible protease sites (*Roudiak and Shrader, 1998*; *Duman and Löwe, 2010*; *Patterson et al., 2004*; *Vasilyeva et al., 2002*). This notion is also corroborated by our high-resolution NMR analyses on MtaLonA NTD both in isolation and as a part of the full-length protein, demonstrating that the independently fast tumbling motion of the NTDs in the ~0.5 MDa hexameric assembly of LonA. Moreover, their highly favorable NMR relaxation properties have allowed us to analyze the interactions of the NTD with unfolded proteins, protein aggregates, degron tags, and intrinsically disordered substrates. Our work provides atomic details of the NTD-mediated substrate discrimination by temperature-based switching of the native folded protein to its unfolded state in an NMR sample. NMR characterization indicates MtaLonA NTD does not interact with well-folded Ig2, native lysozyme, or intrinsically disordered α-casein, whereas unfolded proteins, aggregates, and degron tags can elicit pronounced chemical shift changes to the N-lobe of its NTD, which is subsequently confirmed by structure-based mutagenesis. Collectively, these results suggest that LonA has two substrate-interacting modes: (1) the NTD-non-requiring mode for intrinsically disordered substrates lacking hydrophobic-rich region, which may be engaged directly with the pore-loops in the LonA assembly without involving the six NTDs; (2) the NTD-requiring mode for recognizing and trapping damaged/denatured substrates with exposed hydrophobic regions before their engagement with the pore-loops of the LonA assembly. The key role of NTD is to enable LonA to perform protein quality control to selectively capture and degrade proteins in damaged unfolded states.

Compared with two-lobe organization of LonA NTDs, bacterial FtsH proteases have compact N domains formed by a topology of β1-α1-β2-β3-β4-α2-β5, while ClpXP and ClpEP proteases include N-terminal Zinc-binding domains. ClpAP and ClpCP proteases, as well as of ClpB chaperones, consist of globular α-helical N domains which bear striking resemblances to the C-lobe of MtaLonA NTD (*Rotanova et al., 2019*). All these N domains, although structurally very distinct, are thought to mediate the interactions with substrates or adaptor proteins. The periplasmic N-domain of FtsH contributes to oligomerization and is essential for modulation the activity of the hexamer in conjunction with the membrane proteins HflK and Hfl (*Scharfenberg et al., 2015*). The N-domain of ClpX is required for recognition of adaptors and some substrates (*Baker and Sauer, 2012*). The role of ClpB NTD in protein disaggregation is well characterized by NMR spectroscopy (*Rosenzweig et al., 2015*), demonstrating that ClpB chaperone recognizes exposed hydrophobic stretches in unfolded or aggregated client proteins via a substrate-binding groove in its NTD. In the presence of ClpB NTD, the stability of client proteins can be seriously affected, indicating that the binding of client proteins to ClpB NTD selectively stabilizes their unfolded conformations. Interestingly, intrinsically disordered α-casein can significantly interact with ClpB NTD, but not MtaLonA NTD. Taken together, the α-helical NTD of ClpB chaperones and the beta-stranded N-lobe of MtaLonA NTD comprise substrate-binding sites that play similar roles in specifically recognizing exposed hydrophobic stretches in unfolded or aggregated proteins.

By analyzing the NTD of MtaLonA interactions with damaged Ig2, the scrambled lysozyme, and Sul20 peptide, we conclude that these binding events are mediated by two hydrophobic patches, which comprise (1) L10, V14, I15, P22, V23, and M85 (termed hpI in the following); (2) M75, L77, P78, and L82 (termed hpII) (*Figure 6A*). Our results also demonstrate that a double mutant NN206* (P22 and M85 replaced by Alanine), resulted in decreased hydrophobicity of hpI, has a significant effect on the ability of MtaLonA NTD to bind damaged proteins, indicating that the hydrophobic patches can be essential for substrate recognition and discrimination. We also examine the exposed hydrophobic residues of β-sheet β2/β5/β4, loop L1, and loop L3 located at the N-lobes of other reported structures of LonA NTDs (*Li et al., 2005*; *Duman and Löwe, 2010*; *Li et al., 2010*;

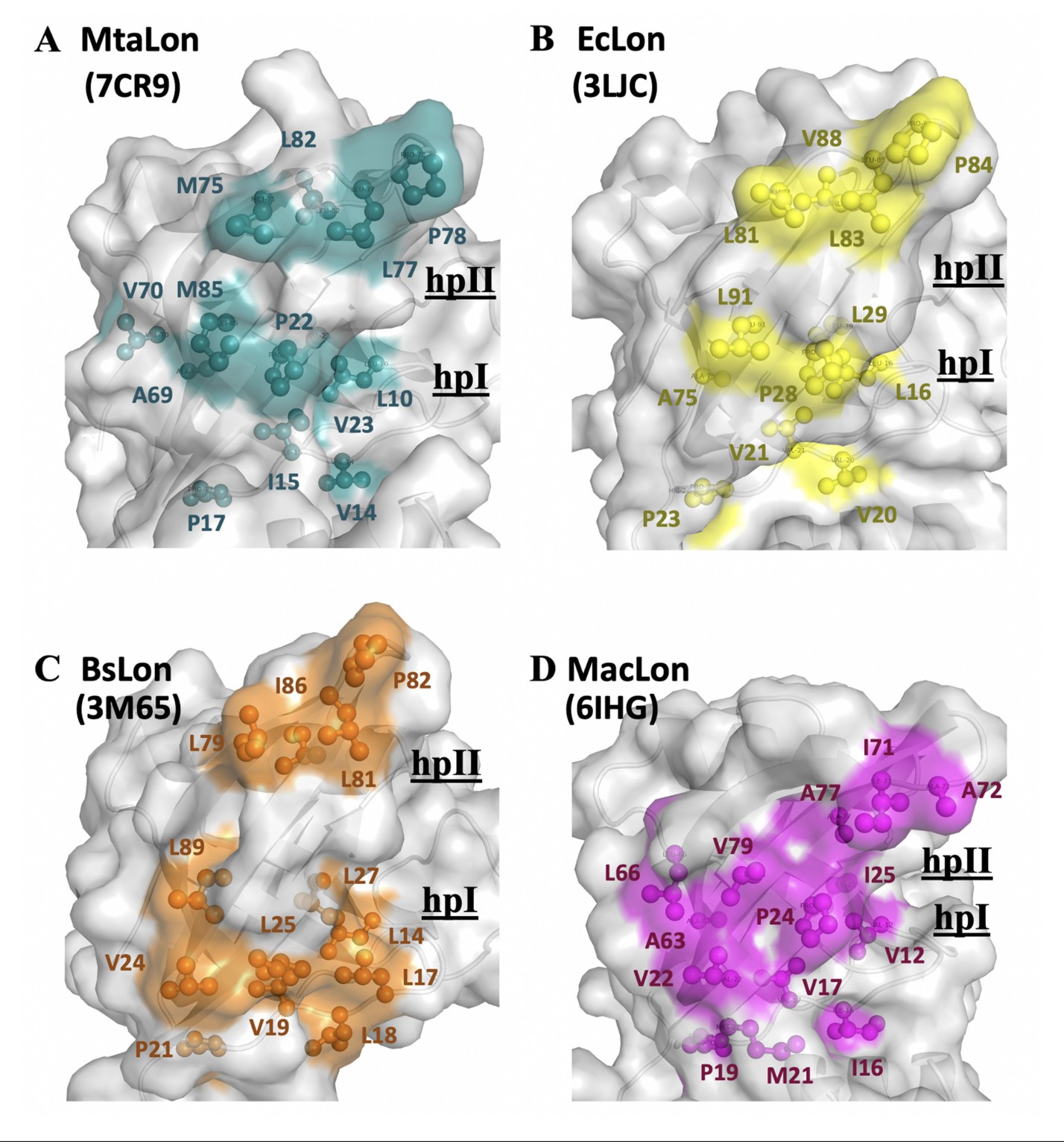

**Figure 6.** The NTD of MtaLonA selectively interacts with unfolded proteins, protein aggregates, and degron-tagged proteins via two hydrophobic patches of its N-lobe. (**A**) Based on the titration data, two hydrophobic patches at the N-lobe of MtaLonA are defined as follows: (1) L10, V14, I15, P22, V23, and M85 (termed hpI in the following); (2) M75, L77, P78, and L82 (termed hpII). (**B**) The hydrophobic patches at the N-lobes of EcLonA NTD are also identified as follows: (1) hpI: L16, V20, V21, P28, L29, and L91; (2) hpII: L81, L83, P84, and V88. (**C**) The Hp I and II of BsLonA are shown as follows: (1) hpI: L14, L17, L18, V19, V24, L25, L27, and L89; (2) hpII: L79, L81, P82, and I86. (**D**) Merged Hp I and II of MacLonA are shown as follows: (1) hpI: V12,

*Figure 6 continued on next page*

*Figure 6 continued*

I16, V17, V22, P24, I25, L66, and V79; (2) hpII: I71, A72, and A77. The conserved hydrophobic patches at the N-lobes may be potentially responsible for conferring substrate selectivity toward damaged proteins and degrons. The PDB codes are indicated in parentheses.

The online version of this article includes the following figure supplement(s) for figure 6:

**Figure supplement 1.** Structural superposition of the N-terminal fragments from MtaLon, EcLon, MacLon, and BsLon.

*Chen et al., 2019*). The N-terminal fragment from *B. subtilis* LonA was previously reported to adopt a domain-swapped dimer where the N-lobe of one monomer is positioned next to the C-lobe of the other monomer (*Duman and Löwe, 2010*). In contrast, the structures of the N-terminal fragments from MtaLon, EcLon, and MacLon show that the N- and C-lobes are joined together via a short linker (*Figure 6—figure supplement 1A,B*). Based on structural alignments (*Figure 6—figure supplement 1C*), the N-lobe of EcLon has almost the same pattern of exposed hydrophobic residues as that of MtaLon NTD (*Figure 6B*). Interestingly, the NTD of MacLon shows merged hp I and II, while the hpI of BsLon NTD is a little distant from hpII (*Figure 6C,D*). The structural comparison reveals that, similarly to MtaLon NTD, all three reported structures exhibit exposed hp I and II with slight variations in their shapes and sizes, suggesting that the conserved hydrophobic patches at the N-lobes may be potentially responsible for conferring substrate selectivity toward damaged proteins and degrons.

It has previously been known that LonA can recognize a degron-tagged protein by binding to its largely exposed hydrophobic residues. Here, our results show the C-terminal Sul20 has substantially low heteronuclear NOE values, reflecting this region undergoes rapid local motion and is solvent exposed in solution. Therefore, MtaLonA NTD can easily recognize and interact with the C-terminal aromatic and hydrophobic residues of Sul20. This finding indicates that a pair of aromatic residues consisting of Y19 and H20 in Sul20 may play crucial roles in the hydrophobic interactions with residues P22 and M85 of MtaLonA NTD, which is verified with a double-mutant NN206*. The analysis of the interaction of NN206 or NN206* with Ig2D5-S20 or Ig2D5-S20-Y19A/H20A suggests that the exposed substrate regions with aromatic or hydrophobic side chains are main determinants for their engagement with NN206, which forms a binding site with similar hydrophobic residues. The primarily hydrophobic or aromatic interactions with NN206, however, do not lead to specific disorder-to-order transition of the substrate, forming for example an extended or helical structures in the bound state. Perhaps, specific backbone or side-chain interactions are not involved in substrate binding by NN206. Rather, the N-lobe of NN206 engages non-specific interactions with non-contiguous patches of exposed hydrophobic residues of the substrates, thereby serving a role mainly to trap the substrates and keep them to the proximity of the AAA+ ring. Furthermore, by selectively binding to hydrophobic residues exposed in the thermally unfolded states of proteins, the NTDs prominently stabilize substrate's unfolded conformation. These results show that the NTD–substrate interactions involve the hydrophobic residues on both partners, suggesting aromatic and hydrophobic side chains, which are usually buried in the native proteins, are important for the NTD to enable MtaLonA to selectively capture and degrade proteins in a damaged unfolded state or with degron tags.

Our results directly demonstrate that thermally damaged proteins and degrons induce chemical shift changes and broadened resonances located mainly in the two hydrophobic patches of N-terminal lobe, but we do not find evidence that the C-lobe of the NTD is involved in substrate interaction. However, upon titration of thermally unfolded Ig2 into U-$^2$H/$^{15}$N-labelled full-length LonA*, affected residues were shown in both N- and C-lobes of MtaLonA* NTD at 55°C (*Figure 7A and B*). Compared $^1$H–$^{15}$N TROSY-HSQC spectra of protonated NN206 and highly deuterated MtaLonA* in the presence of thermally unfolded protein Ig2, the signals from the C-lobe of MtaLonA* NTD showed notable line broadening, suggesting that other NTDs of LonA* hexamer may join to interact with substrates. It is likely that when a substrate like Ig2 becomes damaged or partially unfolded one or more Lon binding sites are exposed. The multiple NTDs in the ~0.5 MDa hexameric assembly of LonA can work as team players and show the avidity effect on substrate binding. When one NTD of LonA hexamer binds to and tethers a substrate nearby, it would be easier for other NTDs to make further substrate contacts. All together, these results reveal the free molecular tumbling of the NTD and its substrate-binding site, based on which a model involving multi-NTD interactions, may be proposed for substrate recognition. The flexibly linked NTD of the hexameric LonA is

swaying from side to side and back and forth to survey, recognize, and trap substrates with exposed hydrophobic sequences (*Figure 7C*). After initial binding, other NTDs may join to increase the avidity of the LonA–substrate interaction (*Figure 7D*); the engagement of a substrate protein by multiple NTDs also serves to facilitate effective substrate unfolding and translocation mediated by the coordinated movements of the pore-loops in the ATPase modules of the LonA chamber, powered by cycles of ATP binding and hydrolysis. Finally, the substrate polypeptide chain is pulled inside the chamber and undergoes proteolysis by protease modules of LonA.

# Materials and methods

**Key resources table**

| Reagent type (species) or resource | Designation | Source or reference | Identifiers | Additional information |
|---|---|---|---|---|
| Peptide | Sul20 peptide (Sequence: ASSHATRQLS GLKIHSNLYH) | GenScript (https://www. genscript.com/) | | at > 95% purity |
| Strain, strain background (*Escherichia coli*) | BL21(DE3) | Novagen | | |
| Plasmid | pET-NN206 | This paper | | |
| Plasmid | pET-NN243 | This paper | | |
| Plasmid | pET-NN291 | This paper | | |
| Plasmid | pET-MtaLon | Reference 18 | | |
| Plasmid | pET-AAAP | Reference 18 | | |
| Plasmid | pET-MtaLon* | Reference 18 | | |
| Plasmid | pET-AAAP* | Reference 18 | | |
| Plasmid | pET-NN206* | This paper | | |
| Plasmid | pET-Ig2 | This paper | | |
| Plasmid | pET-Ig2D5 | This paper | | |
| Plasmid | pET-Ig2D5-S20 | This paper | | |
| Plasmid | pET-Ig2D5-S20-Y19A/H20A | This paper | | A Y19A/H20A double mutant in Ig2D5-S20 construct |
| Software, algorithm | PyMOL | https://pymol. org/2/ | RRID:SCR_000305 | |
| Software, algorithm | CARA | http://cara.nmr. ch/doku.php | | |
| Software, algorithm | NMRPipe | https://www. ibbr.umd. edu/nmrpipe/ install.html | | |
| Software, algorithm | NMRView | http://www. onemoonscientific. com | | |
| Sequence-based reagent | Ig2D5-Sul20 Y19AH20A_F | This paper | PCR primers | TGGTGCTCG AGTTAGGCGGCC AGGTTAGAGTGGAT |
| Sequence-based reagent | Ig2D5-Sul20 Y19AH20A_R | This paper | PCR primers | ATCCACTCTAACCT GGCCGCCTAA CTCGAGCACCA |
| Sequence-based reagent | NN206 P22A_F | This paper | PCR primers | CCACGTCCACCG CGGTGGTGGTGTG |
| Sequence-based reagent | NN206 P22A_R | This paper | PCR primers | CACACCACCAC CGCGGTGGACGTGG |

*Continued on next page*

*Continued*

| Reagent type (species) or resource | Designation | Source or reference | Identifiers | Additional information |
|---|---|---|---|---|
| Sequence-based reagent | NN206 M85A_F | This paper | PCR primers | CCGGGCTTCCACC GCGACCTGCAGGGTG |
| Sequence-based reagent | NN206 M85A_R | This paper | PCR primers | CACCCTGCAGGT CGCGGTGGAA GCCCGG |

## Peptide

Sul20 peptide (sequence: ASSHATRQLSGLKIHSNLYH) was synthesized by GenScript (https://www.genscript.com/) at >95% purity.

## Cloning, protein expression, and purification

The plasmids expressing for full-length MtaLonA (1–793 residues) or AAAP (242–793 residues) with a C-terminal 6xHis-tag were transformed into *E. coli* BL21(DE3) cells (*Su et al., 2016*). Site-directed mutagenesis was performed using the Quickchange kit (Stratagene). NN206 (1–206 residues), NN243 (1–243 residues), and NN291 (1–291 residues) were cloned into pET-modified vector with a tobacco etch virus (TEV) cleavage site. The three resultant plasmids, encoding the proteins NN206, NN243, and NN291 with the N-terminal His-tag, were transformed into *E. coli* BL21(DE3) cells. The target proteins synthesis was induced with 0.5 mM isopropyl-thio-β-D-galactoside (IPTG) at an absorbance at 600 nm ($OD_{600}$) ~0.6 at 28°C. The target proteins were purified with nickel-chelating resins (Ni-NTA, Qiagen). The protein samples were collected and purified by a Superdex 200 (GE Healthcare) column (*Lin et al., 2016*). MtaLonA and AAAP proteins were collected and further purified on a MonoQ (GE Healthcare) chromatography. For NN206, NN243, and NN291 purification, the N-terminal His-tagged proteins were cleaved by TEV protease for overnight at 4°C and then reloaded onto Ni-NTA to remove TEV protease. The flow-through fraction containing target proteins were collected and purified with Superdex 75 (GE Healthcare) chromatography. Ig2 (domains 5 and 6 of the gelation factor ABP-120 of *D. discoideum*) (*McCoy et al., 1999*) and Ig2D5-S20 (Ig2 fused with Sul20) were cloned into pET28a(+)tev for generation of pET28a(+)tev-Ig2 and pET28a(+)tev-Ig2D5-S20. The recombinant protein was induced by 0.5 mM IPTG at $OD_{600}$ ~ 0.6 for 16 hr. The purifications of Ig2, Ig2D5, and Ig2D5-20 is the same as NN206, NN243, and NN291. Isotopically labeled samples for NMR studies were prepared by growing the cells in minimal (M9) medium. All NMR samples (NN206, NN206*, NN243, Ig2, Ig2D5, and Ig2D5_S20) used in this study were protonated except the sample of the full-length MtaLonA was highly deuterated. The samples of full-length MtaLonA for NMR experiments were prepared by supplementing the growing medium with 1 g/l of $^{15}NH4Cl$ and 4 g/l of $^2H_7/^{12}C_6$-glucose in 99.8% $^2H_2O$ (Sigma-Aldrich).

## Substrate degradation assays

Ig2 and α-Casein (Sigma, USA) were used as the substrates in these assays (*Lin et al., 2016*). Four micromolar substrate proteins were incubated with 0.4 µM MtaLonA (hexamer) or AAAP or mutations at 55°C. The degradation reactions were halted by protein sample dye and heat inactivation at 95°C for 5 mins. The treated reaction mixtures were then analyzed by SDS–PAGE.

## Protein crystallization

Crystallizations of NN206 and NN291 were performed at 295 K by hanging drop vapor-diffusion method. For crystallization of NN206, in situ proteolysis was performed (*Wernimont and Edwards, 2009*). One microliter of MtaLonA (10 mg/ml) plus trypsin in ratio 1000:1 was mixed with 1 µl of 0.1 M Tris–HCl (pH 7.5), 0.2 M sodium acetate, and 30% PEG 4000. The NN206 crystals appeared in the mixed drop after 3 days. The crystals of NN291 were grown by mixing with 1 µl of NN291 protein sample (15 mg/ml) and 1 µl of well solution containing 0.1 M MES (pH 5.8) and 0.8 M ammonium sulfate. Crystals grew to 0.2–0.3 mm over 2 weeks. Crystals of NN206 and NN291 were cryoprotected with 20% glycerol or 20% ethylene glycol before data collection.

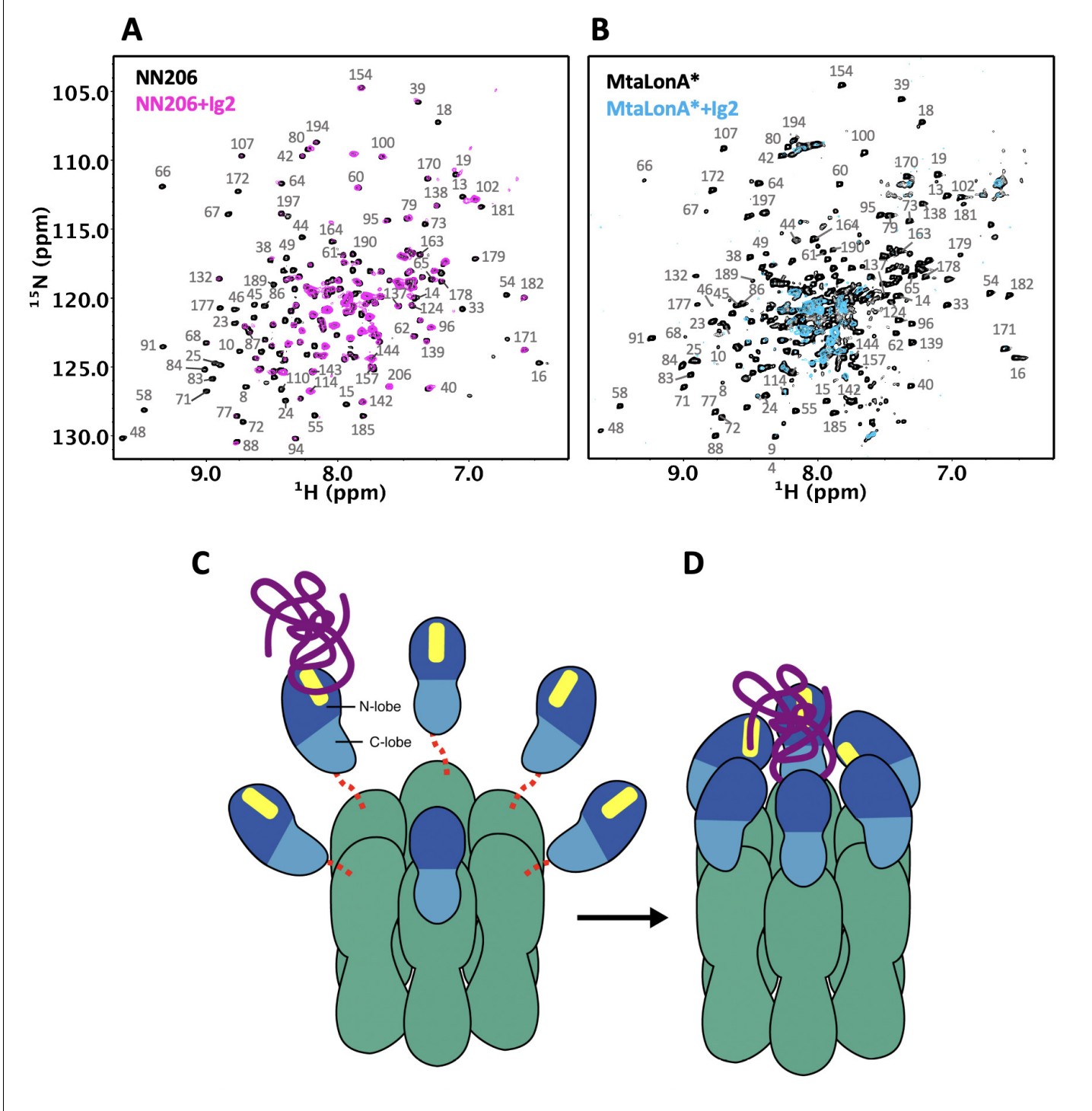

**Figure 7.** Proposed model for NTD-mediated substrate interaction. (**A**) $^1H$–$^{15}N$ TROSY-HSQC spectra of protonated $^{15}N$-labeled NTD in the absence (black) and presence (magenta) of unlabeled Ig2 recorded at 55°C. (**B**) $^1H$–$^{15}N$ TROSY-HSQC spectra of highly deuterated $^{15}N$-labled LonA* in the absence (black) and presence (blue) of unlabeled Ig2 recorded at 55°C. (**C**) The flexibly linked NTD of the hexameric LonA is swaying from side to side and back and forth to survey, recognize, and trap substrates with exposed hydrophobic sequences. Substrate, N-lobe, C-lobe, and the ATPase– Protease chamber are illustrated in purple, marine, blue, and green colors, respectively. The substrate-binding patches are depicted in yellow. Dashed lines represent the flexible linker regions. (**D**) After initial binding, other NTDs may join to increase the avidity of the LonA–substrate interaction. The substrate polypeptide chain is pulled inside the chamber and undergoes proteolysis by protease modules of LonA.

## Structure determination

The data sets of NN206 and NN291 were collected at the beamlines BL-13C1 of National Synchrotron Radiation Research Center (Taiwan) and BL-1A of Photon Factory (Japan), respectively. All data sets were processed by HKL2000 (*Otwinowski and Minor, 1997*). The structure of NN206 was solved by molecular replacement with the program Phaser (*McCoy, 2007*) using the structures of the N-terminal fragment of *B. subtilis* LonA (residues 4–116, PDB code 3M65) and the N-terminal fragment of *E. coli* LonA (residues 120–210, PDB code 3LJC) as the search models. The structure of NN291 was determined by molecular replacement using NN206 structure. All initial models were automatically rebuilt using the program AutoBuild (*Terwilliger et al., 2008*). Subsequently, these structures were refined by manual refitting in Coot (*Emsley and Cowtan, 2004*) and performance of refinement using the program Refmac5 (*Murshudov et al., 2011*). Crystallographic and refinement statistics are listed in *Table 1*. All structure figures were made using PyMOL (Version 1.3, Schrödinger, LLC). The atomic coordinates and structure factors for NN206 and NN291 have been deposited in the Protein Data Bank (http://www.rcsb.org/) with the accession numbers 7CR9 and 7CRA, respectively.

## Thermal shift assay

Thermal shift assay was carried out in qPCR 8-strip tubes (Gunster Biotech) using SYBRO orange (Life Technologies) as dye. For each reaction, 5 µM purified MtaLonA, NN206, or Ig2 was mixed with assay buffer (50 mM NaPi, 2 mM β-Me, pH 6.5) and SYPRO Orange dye (5× final concentration) in 45 µl total volume/well. One hundred micromolar Ig2D5 was mixed with assay buffer and SYPRO Orange dye (10× final concentration) in 45 µl total volume/well. Samples were heated at a rate of 0.5°C per minute from 25°C to 95°C, and fluorescence signals were recorded with the CFX Connect Real-Time PCR Detection System (Bio-Rad). The melting temperatures were analyzed using the derivative plots of the melting curve.

## NMR spectroscopy

NMR experiments were acquired on Bruker 800-MHz spectrometers (Bruker BioSpin, Karlsruhe, Germany). The assignments of NN206 and Ig2D5-S20 backbone $^{15}$N, $^{1}$H$^{N}$, $^{13}$Cα, $^{13}$Cβ, and $^{13}$C′ chemical shifts were obtained by non-TROSY versions of $^{15}$N-$^{1}$H HSQC, HNCACB, HN(CO)CACB, HNCA, HN(CO)CA, and HNCO spectra (*Sattler et al., 1999*). NMR samples containing 300 µM [U-$^{15}$N]-labeled protonated NN206 in the absence and presence of 600 µM protonated Ig2 are prepared in the buffer: 50 mM sodium phosphate (pH 6.5) with 5% D$_2$O. For temperature cycling experiments, the first 2D $^{1}$H–$^{15}$N TROSY-HSQC spectrum was acquired at 32°C; the same sample was heated to 55°C for recording the second spectra, and the temperature was decreased to 32°C for recording the third spectra. A significance level for CSP set by the average value plus one standard deviation and manual inspection of the affected residues establishes if this level needed further adjustment.

For characterization of substrate conformation selection by MtaLonA NTD, 2D $^{1}$H-$^{15}$N TROSY-HSQC NMR spectra of 50 µM [U-$^{15}$N]-labeled protonated Ig2D5 titrated with 50 µM unlabeled protonated NN206, MtaLonA, or AAAP proteins were recorded during a thermal cycle starting from 32°C to 60°C, and then back to 32°C. Seventy-two residues of Ig2D5 were selected for calculating folded ratio. The interaction of NN206 with Sul20 peptide was recorded with 50 µM [U-$^{15}$N]-labeled NN206 titrated with unlabeled 50 and 150 µM unlabeled Sul20 peptide. The interaction of NN206 with TCEP-treated denatured lysozyme was recorded with 50 µM [U-$^{15}$N]-labeled protonated NN206 titrated with 25 µM unlabeled denatured lysozyme. $^{1}$H-$^{15}$N NOE data were recorded in an interleaved manner: one spectrum with 4 s recycle delay followed by 4 s saturation and another spectrum with no saturation and 8 s recycle delay. All NMR data were processed and analyzed by XWIN-NMR (Bruker BioSpin), NMRPipe (*Delaglio et al., 1995*), and NMRView (*Downing, 2004*).

## Acknowledgements

We thank Dr. Shang-Te Danny Hsu for helpful discussion. We also thank Ms S-L Huang of Ministry of Science and Technology (National Taiwan University) for the assistance in NMR experiments. We are grateful to the staff of the Biomedical Resource Core at the First Core Labs, National Taiwan University College of Medicine, for technical assistance. Portions of this research were carried out at

beamlines 13B1 and 13C1 of the National Synchrotron Radiation Research Center (Taiwan). This work was supported by National Taiwan University Grants 108L7809 and 109L7809; and the Ministry of Science and Technology, Taiwan (grant numbers: MOST108-2113-M-002–006- and MOST109-2113-M-002–018- to S-RT and MOST105-2320-B-001–015-MY3 to C-IC).

## Additional information

### Funding

| Funder | Grant reference number | Author |
|---|---|---|
| Ministry of Science and Technology, Taiwan | MOST108-2113-M-002-006- | Shiou-Ru Tzeng |
| Ministry of Science and Technology, Taiwan | MOST109-2113-M-002-018- | Shiou-Ru Tzeng |
| Ministry of Science and Technology, Taiwan | MOST105-2320-B-001-015-MY3 | Chung-I Chang |
| National Taiwan University | 108L7809 | Shiou-Ru Tzeng |
| National Taiwan University | 109L7809 | Shiou-Ru Tzeng |

The funders had no role in study design, data collection and interpretation, or the decision to submit the work for publication.

### Author contributions

Shiou-Ru Tzeng, Conceptualization, Data curation, Formal analysis, Funding acquisition, Investigation, Methodology, Writing - original draft, Project administration, Writing - review and editing; Yin-Chu Tseng, Chia-Ying Hsu, Data curation, Formal analysis; Chien-Chu Lin, Data curation, Formal analysis, Writing - original draft; Shing-Jong Huang, Yi-Ting Kuo, Data curation; Chung-I Chang, Conceptualization, Data curation, Formal analysis, Funding acquisition, Investigation, Writing - original draft, Project administration, Writing - review and editing

### Author ORCIDs

Shiou-Ru Tzeng (iD) https://orcid.org/0000-0002-7180-0190
Chung-I Chang (iD) https://orcid.org/0000-0003-0989-1228

### Decision letter and Author response

Decision letter https://doi.org/10.7554/eLife.64056.sa1
Author response https://doi.org/10.7554/eLife.64056.sa2

## Additional files

### Supplementary files

• Transparent reporting form

### Data availability

Diffraction data have been deposited in PDB under the accession codes 7CR9 and 7CRA. NMR chemical shifts have been deposited in the BioMagResBank: BMRB ID 50697 for NN206, 50733 for NN206*, 50735 for NN206 in Sul20-bound state, 50698 for IgD5, 50702 for IgD5-Sul20 and 50736 for IgD5-Sul20 in MtaNTD-bound state.

The following datasets were generated:

| Author(s) | Year | Dataset title | Dataset URL | Database and Identifier |
|---|---|---|---|---|
| Lin CC, Chang CI | 2020 | Molecular insights into substrate recognition and discrimination by the N-terminal domain of Lon AAA | https://www.rcsb.org/structure/unreleased/7CR9 | RCSB Protein Data Bank, 7CR9 |

| | | | | |
|---|---|---|---|---|
| | | + protease | | |
| Lin CC, Chang CI | 2020 | Molecular insights into substrate recognition and discrimination by the N-terminal domain of Lon AAA + protease | https://www.rcsb.org/structure/unreleased/7CRA | RCSB Protein Data Bank, 7CRA |
| Tzeng SR | 2021 | Molecular insights into substrate recognition and discrimination by the N-terminal domain of Lon AAA + protease | http://www.bmrb.wisc.edu/data_library/summary/index.php?bmrbId=50697 | Biological Magnetic Resonance Data Bank, 50697 |
| Tzeng SR | 2021 | Molecular insights into substrate recognition and discrimination by the N-terminal domain of Lon AAA + protease | http://www.bmrb.wisc.edu/data_library/summary/index.php?bmrbId=50733 | Biological Magnetic Resonance Data Bank, 50733 |
| Tzeng SR | 2021 | Molecular insights into substrate recognition and discrimination by the N-terminal domain of Lon AAA + protease | http://www.bmrb.wisc.edu/data_library/summary/index.php?bmrbId=50698 | Biological Magnetic Resonance Data Bank, 50698 |
| Tzeng SR | 2021 | Molecular insights into substrate recognition and discrimination by the N-terminal domain of Lon AAA + protease | http://www.bmrb.wisc.edu/data_library/summary/index.php?bmrbId=50702 | Biological Magnetic Resonance Data Bank, 50702 |
| Tzeng SR | 2021 | Molecular insights into substrate recognition and discrimination by the N-terminal domain of Lon AAA + protease | http://www.bmrb.wisc.edu/data_library/summary/index.php?bmrbId=50735 | Biological Magnetic Resonance Data Bank, 50735 |
| Tzeng SR | 2021 | Molecular insights into substrate recognition and discrimination by the N-terminal domain of Lon AAA + protease | http://www.bmrb.wisc.edu/data_library/summary/index.php?bmrbId=50736 | Biological Magnetic Resonance Data Bank, 50736 |

The following previously published datasets were used:

| Author(s) | Year | Dataset title | Dataset URL | Database and Identifier |
|---|---|---|---|---|
| Li M, Rasulova F, Melnikov EE, Rotanova TV, Gustchina A, Maurizi MR, Wlodawer A | 2005 | Crystal structure of the N-terminal domain of E. coli Lon protease | https://www.rcsb.org/structure/2ANE | RCSB Protein Data Bank, 2ANE |
| Duman RE, Lowe J | 2010 | Crystal structures of Bacillus subtilis Lon protease | https://www.rcsb.org/structure/3M65 | RCSB Protein Data Bank, 3M65 |
| Li M, Gustchina A, Rasulova FS, Melnikov EE, Maurizi MR, Rotanova TV, Dauter Z, Wlodawer A | 2010 | Structure of the N-terminal fragment of Escherichia coli Lon protease | https://www.rcsb.org/structure/3LJC | RCSB Protein Data Bank, 3LJC |
| Chen X, Zhang S, Bi F, Guo C, Feng L, Wang H, Yao H, Lin D | 2019 | Crystal structure of the N domain of Lon protease from Mycobacterium avium complex. | https://www.rcsb.org/structure/6IHG | RCSB Protein Data Bank, 6IHG |
| McCoy AJ, Fucini P, Noegel AA, Stewart M | 2000 | Structural basis for dimerization of the Dictyostelium gelation factor (ABP120) rod | https://www.rcsb.org/structure/1QFH | RCSB Protein Data Bank, 1QFH |

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
