## [Decision Letter]

**Acceptance summary:**

In this study, Tzeng et al. provide molecular insights into the mode of substrate recognition by the N-terminal flexible domain of the ubiquitous AAA+ protease. The authors combine atomic resolution structural information, from NMR spectroscopy, together with functional data to show that LonA selectively interacts with unfolded proteins, protein aggregates, and degron-tagged proteins through two hydrophobic patches located in its N-terminal domain. Furthermore, the authors show that the NTD binds to hydrophobic regions exposed only in the thermally unfolded states of substrate proteins, with this binding blocking their refolding. These findings suggest that LonA selectively recognizes damaged and misfolded proteins via the interaction of their exposed hydrophobic residues with LonA N-terminal domains.

**Decision letter after peer review:**

Thank you for submitting your article "Molecular insights into substrate recognition and discrimination by the N-terminal domain of Lon AAA+ protease" for consideration by *eLife*. Your article has been reviewed by three peer reviewers, and the evaluation has been overseen by Rina Rosenzweig as a Reviewing Editor and David Ron as the Senior Editor. The following individuals involved in review of your submission have agreed to reveal their identity: Björn Burmann (Reviewer #1); Mark Foster (Reviewer #3).

The reviewers have discussed the reviews with one another and the Reviewing Editor has drafted this decision to help you prepare a revised submission.

Summary:

In this study, Tzeng et al. provide molecular insights into the mode of substrate recognition by the N-terminal flexible domain of the ubiquitous AAA+ protease. The authors combine atomic resolution structural information from NMR spectroscopy together with functional data to show that LonA selectively interacts with unfolded proteins, protein aggregates, and degron-tagged proteins by two hydrophobic patches located in its N-terminal domain.

Furthermore, the authors show that the NTD binds to hydrophobic regions exposed only in the thermally unfolded states of substrate proteins, with this binding blocking their refolding. These findings suggest that LonA selectively recognizes damaged and misfolded proteins via the interaction of their exposed hydrophobic residues with LonA N-terminal domains.

Overall, all reviewers agree that the results presented in this paper are of great interest and advance the field of protein quality control by AAA+ machineries. However, while the majority of the authors' conclusions are well supported by the presented data, the reviewers feel that the following points must be addressed prior to considering this manuscript for a publication in *eLife*.

Essential revisions:

1) The majority of the NMR data presented in this manuscript is qualitative. In order to support the conclusion of the paper, quantitative analysis of the NMR data should also be provided.

This, would include chemical shift perturbation and/or intensity ratio plots for 15N NN206 + Ig2 (Figure 3), 15N NN206* mutant + Ig2 (Figure 3), 15N NN206 + Sul20 (Figure 4), and 15N Ig2D5-S20 + NN206 (Figure 5) interactions, as well as for the data presented in supplementary figures Figure 3—figure supplement 2A, S4B, Figure 4—figure supplement 1B and E.

In addition, the authors should provide a comparison between the NMR-derived secondary structure for the NN206 construct and their solved X-ray structure.

Furthermore, an analysis should also be included of the changes to the conformation / secondary structure of Sul20 upon interaction with NN206.

2) It would be helpful for the reader if the Discussion section included a detailed analysis of how the mechanism of LonA NTD substrate recognition and subsequent unfolding, uncovered in this manuscript, compares to other known proteins with similar functions – for example heat shock proteins or other chaperones, chaperonins, or other protein quality control members.

3) In general, the quality of the figures should be improved and the figures need to be remade so they fit into a single page / panel. Some of the material can be moved from the main to the supplementary.

It would be helpful for the readers if a thorough and consistent color coding of the different domains were used. The axis labels for the spectra should also be consistent throughout the figure/panel and should be written in a large enough font to read in the printed version. We would further suggest avoiding the use of red and green side by side in the figures.

4) The backbone resonance assignments should be deposited to the BMRB.

In addition we believe the following two suggestions made by the reviewers could greatly strengthen the conclusions of the paper.

1) The authors very convincingly show the importance of P22 and M85 to substrate recognition by characterizing the catalytic and structural/interaction properties of the P22A/M85A double mutant. Based on NMR spectra, they identify Y19 and H20 of the Sul20 degron as potentially lynchpin residues for the interaction with NN206. Showing that a Y19A/H20A double mutant of Sul20 does not bind the NTD and does not target proteins for degradation by the full-length protease would strengthen this conclusion.

2) The authors propose a model for substrate recognition and binding by the NTD of LonA, based on their new data and available structures. In this model (Figure 7C,D) the substrate is bound by one NTD in an "open" conformation, then is sequestered by binding to multiple NTDs. The model is supported by the observation that when the Ig2 substrate is added to full-length MtaLonA, amide resonances assignable to NTDs are significantly broadened, whereas if the substrate were engaged by a single NTD one would expect to observe the resonances from the other non-engaged NTDs. Although this model would be strengthened by quantitative data on stoichiometry and affinity, the model is certainly consistent with the data.

---

## [Author Response]

Essential revisions:1) The majority of the NMR data presented in this manuscript is qualitative. In order to support the conclusion of the paper, quantitative analysis of the NMR data should also be provided.This, would include chemical shift perturbation and/or intensity ratio plots for 15N NN206 + Ig2 (Figure 3), 15N NN206* mutant + Ig2 (Figure 3), 15N NN206 + Sul20 (Figure 4), and 15N Ig2D5-S20 + NN206 (Figure 5) interactions, as well as for the data presented in supplementary figures Figure 3—figure supplement 2A, S4B, Figure 4—figure supplement 1B and E.In addition, the authors should provide a comparison between the NMR-derived secondary structure for the NN206 construct and their solved X-ray structure.Furthermore, an analysis should also be included of the changes to the conformation / secondary structure of Sul20 upon interaction with NN206.

We thank the reviewer for bringing out these points. As suggested, we have added results of the quantitative analysis to the supplementary data.

We have added quantitative analysis of the NMR data to the supplementary, including:

^15^N NN206 + Ig2 (Figure 3—figure supplement 4A)

^15^N NN206* mutant + Ig2 (Figure 3—figure supplement 4F)

^15^N NN206 + Sul20 (Figure 4—figure supplement 1A)

^15^N NN206 + casein (Figure 3—figure supplement 2B)

^15^N NN206 vs^15^N NN206* mutant (Figure 3—figure supplement 4E)

^15^N NN206* + Sul20 (Figure 4—figure supplement 1C)

^15^N Ig2D5-S20 + NN206 (Figure 4F and Figure 5—figure supplement 1D)

^15^N Ig2D5-S20 + NN206* (Figure 4—figure supplement 1F and Figure 5—figure supplement 2D)

Globally, the NMR-derived secondary structure of NN206, based on the method of Chemical Shift Index, is highly similar to its solved X-ray structure. A comparison between the NMR-derived secondary structure for the NN206 construct and the solved X-ray structure is added to Figure 2—figure supplement 1D.

Also, the NMR-derived secondary structure of Sul20 upon interaction with NN206 is added to Figure 4—figure supplement 1D. The secondary structure of Sul20 shows random coil and it remains unchanged in the presence of the NTD of MtaLonA.

2) It would be helpful for the reader if the Discussion section included a detailed analysis of how the mechanism of LonA NTD substrate recognition and subsequent unfolding, uncovered in this manuscript, compares to other known proteins with similar functions – for example heat shock proteins or other chaperones, chaperonins, or other protein quality control members.

We thank the reviewer suggestion and have added a discussion accordingly:

“Compared with two‐lobe organization of LonA NTDs, bacterial FtsH proteases have compact N domains formed by a topology of β1-α1-β2-β3-β4-α2-β5 while ClpXP and ClpEP proteases include N-terminal Zinc-binding domains. ClpAP and ClpCP proteases, as well as of ClpB chaperones, consist of globular α‐helical N domains which bear striking resemblances to the C-lobe of MtaLonA NTD. All these N domains, although structurally very distinct, are thought to mediate the interactions with substrates or adaptor proteins. The periplasmic N-domain of FtsH contributes to oligomerization and is essential for modulation the activity of hexamer in conjunction with the membrane proteins HflK and Hfl. The N-domain of ClpX is required for recognition of adaptors and some substrates. The role of ClpB NTD in protein disaggregation is well characterized by NMR spectroscopy, demonstrating that ClpB chaperone recognizes exposed hydrophobic stretches in unfolded or aggregated client proteins via a substrate-binding groove in its NTD. In the presence of ClpB NTD, the stability of client proteins was seriously affected, indicating that the binding of client proteins to ClpB NTD selectively stabilizes their unfolded conformations. Interestingly, intrinsically disordered α-casein can significantly interact with ClpB NTD but not MtaLonA NTD. Taken together, the α-helical NTD of ClpB chaperone and the β-stranded N-lobe of MtaLonA NTD comprise substrate-binding sites that play similar roles in specifically recognizing exposed hydrophobic stretches in unfolded or aggregated proteins.”

3) In general, the quality of the figures should be improved and the figures need to be remade so they fit into a single page / panel. Some of the material can be moved from the main to the supplementary.It would be helpful for the readers if a thorough and consistent color coding of the different domains were used. The axis labels for the spectra should also be consistent throughout the figure/panel and should be written in a large enough font to read in the printed version. We would further suggest avoiding the use of red and green side by side in the figures.

We thank the reviewer for these comments. The figures are updated with higher resolution. Figures 2E, 3E, and 3G are moved from the main to Figure 2—figure supplement 1A, Figure 3—figure supplement 3 and Figure 3—figure supplement 4B. The axis labels are changed to a large enough font.

4) The backbone resonance assignments should be deposited to the BMRB.

We agree and backbone resonance assignments have been deposited in the Biological Magnetic Resonance Data Bank under the following accession codes:

50697 for NN206, 50733 for NN206*, 50735 for NN206 in sul20-bound state, 50698 for IgD5, 50702 for IgD5-Sul20 and 50736 for IgD5-Sul20 in MtaNTD-bound state.

In addition we believe the following two suggestions made by the reviewers could greatly strengthen the conclusions of the paper.1) The authors very convincingly show the importance of P22 and M85 to substrate recognition by characterizing the catalytic and structural/interaction properties of the P22A/M85A double mutant. Based on NMR spectra, they identify Y19 and H20 of the Sul20 degron as potentially lynchpin residues for the interaction with NN206. Showing that a Y19A/H20A double mutant of Sul20 does not bind the NTD and does not target proteins for degradation by the full-length protease would strengthen this conclusion.

The point is well taken. To address the above points, we generated a Y19A/H20A double mutant in Ig2D5-S20 construct (hereafter abbreviated as Ig2D5-S20-Y19A/H20A) and have added the results in the text as follows:

“To investigate an effect attributed to Y19A/H20A double mutant, a gel-based assay was performed at 42°C while Ig2D5 was still at its native state. The results showed that the degradation by the full-length protease against Y19A/H20A double mutant of Sul20 is much less efficient (Figure 4—figure supplement 2A). The NMR spectra of Ig2D5-S20 and Ig2D5-S20-Y19A/H20A can be well superimposed and only the chemical shifts of residues Y19A and H20A are largely changed (Figure 4—figure supplement 2B and 2C). By addition of ^15^N-labeled NN206 to ^15^N-labeled Ig2D5-S20-Y19A/H20A, careful comparison of chemical shifts reveals that no changes, suggesting that a Y19A/H20A double mutant indeed significantly reduces the binding with NN206 (Figure 4F).”

2) The authors propose a model for substrate recognition and binding by the NTD of LonA, based on their new data and available structures. In this model (Figure 7C,D) the substrate is bound by one NTD in an "open" conformation, then is sequestered by binding to multiple NTDs. The model is supported by the observation that when the Ig2 substrate is added to full-length LonA, amide resonances assignable to NTDs are significantly broadened, whereas if the substrate were engaged by a single NTD one would expect to observe the resonances from the other non-engaged NTDs. Although this model would be strengthened by quantitative data on stoichiometry and affinity, the model is certainly consistent with the data.

We appreciate the reviewer comment and suggestion. It is likely that when a substrate like Ig2 becomes damaged or partially unfolded one or more Lon binding sites are exposed. The multiple NTDs in the ~0.5 MDa hexameric assembly of LonA can work as team players and show avidity effect on substrate bonding. When one NTD of LonA hexamer binds to and tethers a substrate nearby, it would be easier for other NTDs to make further substrate contacts. In fact, we have tried to detect the stoichiometry and affinity for the interaction between unfolded Ig2 and full-length LonA by ITC at 55°C; however, the change of binding enthalpy is too small to characterize the thermodynamics of this interaction. We also had thought an alternate biophysical method (e.g., SPR) to explore the stoichiometry but it is difficult to perform experiments at 55°C.